# Fatigue, Depression and Health-Related Quality of Life in Patients with Post-Myocardial Infarction during the COVID-19 Pandemic: Results from the Augsburg Myocardial Infarction Registry

**DOI:** 10.3390/jcm12196349

**Published:** 2023-10-03

**Authors:** Christa Meisinger, Inge Kirchberger, Philip Raake, Jakob Linseisen, Timo Schmitz

**Affiliations:** 1Epidemiology, Medical Faculty, University of Augsburg, 86156 Augsburg, Germany; inge.kirchberger@med.uni-augsburg.de (I.K.); jakob.linseisen@med.uni-augsburg.de (J.L.); timo.schmitz@med.uni-augsburg.de (T.S.); 2Department of Cardiology, Respiratory Medicine and Intensive Care, University Hospital Augsburg, 86156 Augsburg, Germany; philip.raake@uk-augsburg.de; 3Institute for Medical Information Processing, Biometry and Epidemiology, Ludwig-Maximilians Universität München, 81377 Munich, Germany

**Keywords:** depression, fatigue, health-related quality of life, COVID-19, myocardial infarction

## Abstract

The interplay between fatigue and depression and their association with health-related quality of life (HRQoL) after acute myocardial infarction (AMI) has received little attention during the COVID-19 pandemic. Therefore, this study evaluated the frequency of fatigue and depression in post-AMI patients during the COVID-19 pandemic and investigated the cross-sectional associations between fatigue, depression and HRQoL. Methods: The analysis was based on population-based Myocardial Infarction Registry Augsburg data. All survivors of AMI between 1 June 2020 and 15 September 2021 were included (*n =* 882) and received a postal questionnaire containing questions about fatigue (Fatigue Assessment Scale), depression (Patient Health Questionnaire), and HRQoL (MacNew Heart Disease HRQoL questionnaire) on 17 November 2021. The questionnaire was returned by 592 patients (67.1%), and 574 participants could be included in the analysis. Multivariable linear regression models were performed to investigate the associations between fatigue and depression (both exposures) and HRQoL (outcome). Results: Altogether, 273 (47.6%) participants met the criteria for the presence of fatigue, about 16% showed signs of moderate to severe depression. Both fatigue and depression were significantly associated with a decreased HRQoL (total score and emotional, social, and physical subscales; all *p*-values < 0.0001). In particular, a combined occurrence of fatigue and depression was associated with a significantly reduced HRQoL. Conclusions: It seems necessary to screen post-MI patients for the presence of fatigue and depression in clinical practice on a routine basis to provide them with adequate support and treatment and thus also to improve their HRQoL.

## 1. Introduction

Previous studies have reported that health-related quality of life (HRQoL) may be negatively affected for years after acute myocardial infarction (AMI) [1,2]. Both depression [3] and fatigue [4] are common in AMI patients. They are associated with restrictions in daily physical activity [5,6], lack of motivation as well as feelings of demoralization [7], subsequently leading to a reduced HRQoL in these patients [8,9,10]. Depressive symptoms and reported fatigue in coronary heart disease (CHD) patients [11] and post-AMI patients [12,13] are closely related, as fatigue is one of the main symptoms of depression [14]. Nevertheless, despite the close relationship between fatigue and depression after AMI [15], fatigue symptoms can also occur without concurrent depression [16]. 

The COVID-19 pandemic has resulted in a global crisis that negatively affects physical well-being and mental health. In this context, studies have shown that patients with chronic illnesses had significantly lower HRQoL during the pandemic and were more likely to suffer from depressive symptoms [17]. It could be assumed that patients with chronic conditions may be more vulnerable to the psychosocial effects of a pandemic and associated public health interventions [18]. Furthermore, social deprivation and isolation in patients with chronic disorders may be associated with adverse psychological outcomes and premature mortality [19,20].

So far, the interplay between fatigue and depression and their association with HRQoL after AMI has received little attention and, to our knowledge, has not been investigated, especially during the COVID-19 pandemic. In the context of the COVID-19 pandemic, certain measures may have triggered negative thoughts that led to the onset of depression, anxiety, and fatigue. In addition, possible job loss, trauma, or health threats due to the pandemic and imposed curfews or social distancing may have caused psychological stress and exacerbated existing mental health problems that could interact with cardiovascular disease [21]. It was also found that the COVID-19 pandemic impacted the outcomes of acute coronary syndromes [22] and the quality of life in AMI survivors [23]. Thus, this study aimed to evaluate the frequency of fatigue and depression in post-AMI patients during the COVID-19 pandemic and to investigate the associations between fatigue, depression and HRQoL.

## 2. Materials and Methods

### 2.1. Study Population

The present study was based on data from the population-based Myocardial Infarction Registry Augsburg, which registers all cases of non-fatal AMIs in the study region (city of Augsburg and two adjacent counties, Aichach-Friedberg and Augsburg) in Southern Germany. The myocardial infarction registry was established as part of the World Health Organization (WHO) MONICA (Monitoring Trends and Determinants in Cardiovascular Disease) project in 1984 [24]. MONICA was terminated in 1995, and the registry was subsequently continued within the framework of KORA (Cooperative Health Research in the Augsburg Region). Since 2019, the registry has continued under the Augsburg Myocardial Infarction Registry. Inclusion criteria for the registry are the patient is an inhabitant of the region of Augsburg, is at least 25 years old, and survived 24 h after hospital admission [25]. These inclusion criteria were established as part of the MONICA study and are still used today. For the present study, all cases of AMIs admitted to the hospital between 1 June 2020 and 15 September 2021 and still alive were included (*n =* 882). There were no exclusion criteria. A postal questionnaire was sent to all those patients on 17 November 2021. All patients who did not respond within four weeks (*n =* 504) received a postal reminder on 16 December 2021. Altogether, 592 patients filled in the questionnaire (67.1%).

Meanwhile, 33 patients had died, 16 patients had dementia and could not answer the questions, and eight refused participation. No response was received from 233 patients (26.4%). This study followed the STROBE (Strengthening the Reporting of Observational Studies in Epidemiology) Statements [26].

### 2.2. Ethical Consideration

The study complies with the Declaration of Helsinki. All study participants gave written informed consent, and the study was approved by the Ethics Committee of the Bavarian Medical Association (Bayerische Landesärztekammer; No 12057). 

### 2.3. Data Collection

Baseline data was collected through a face-to-face interview during the patient’s hospital stay at a general ward after discharge from the intensive care unit. In addition, data on treatment in the hospital, medication, comorbidities, etc., were extracted from the medical chart. At the postal follow-up survey, data was collected on paper. The participants were asked about the impact of COVID-19 on their lifestyle, fears and worries in connection with the pandemic, their HRQoL, depression fatigue, and where patients get information on health-related topics.

To assess HRQoL, the MacNew heart disease health-related quality of life questionnaire was used. This questionnaire captures patients’ feelings about how ischemic heart disease affects daily functioning by 27 items with a total HRQoL score and three subscales for physical limitations, emotional functioning, and social functioning [27,28,29]. The items are scored on a scale from 1 (low HRQoL) to 7 (high HRQoL) [30]. The German version of the MacNew was validated in patients with AMI and demonstrated satisfactory reliability, validity, and sensitivity [29,31].

The nine-item Patient Health Questionnaire (PHQ-9) was used to assess depression symptoms [32]. The questionnaire applied a 4-point Likert scale, and participants rated the frequency of symptoms ranging from 0 (Not at all) to 3 (Nearly all days). The PHQ-9 can be used as a sum score to determine the severity of the depressiveness but can also be used categorically. Thus, the scale value “depressiveness” corresponds to the sum of the score and ranges from 0 to 27. For use as a categorical variable, a calculated scale sum value ≥ 10 was applied to define moderate to high severity of depressive symptoms [33]. The German version of the PHQ-9 showed good psychometric properties [34].

Fatigue was assessed using the Fatigue Assessment Scale (FAS), a reliable and valid instrument for assessing fatigue symptoms [35]. This scale consists of 10 items with response options ranging from “never” (1) to “always” (5), and fatigue levels between a minimum of 10 and a maximum of 50 can be calculated. Scores below 22 are regarded as ‘no fatigue’, scores between 22 and 35 are considered ‘moderate fatigue’, and scores above 35 are defined as ‘severe fatigue’ [36]. In the present analysis, individuals with scores of 22 and above were classified as having fatigue.

### 2.4. Statistical Analysis

Continuous variables were described as means ± standard deviations (SD), and categorical variables as absolute and relative frequencies. Differences between the two groups were assessed by *t*-test in the case of continuous variables and Chi-square test in case of categorical variables. After conducting univariable linear regression analyses, multivariable linear regression models were performed to investigate the associations between fatigue and depression (exposures) and the HRQoL (total score and emotional, social, and physical subscales) as an outcome. First, fatigue and depression were each included in the model as categorized variables and based on literature review adjusted for age, sex, diabetes, prior stroke, German nationality, highest school education, living alone, smoking status, BMI, any recanalisation therapy, hypertension, STEMI infarction, employment status, and prior SARS-CoV-2 infection. As covariables, we included in our regression models variables which are associated with the exposure, the outcome or both and which are not mediators [37].

Second, to examine the joint effect of fatigue (yes/no) and depression (moderate to severe vs. no) on the HRQoL after AMI, a combined fatigue and depression variable was generated. Patients were classified into four categories: no fatigue/no depression, no depression/fatigue, moderate to severe depression/no fatigue, and moderate to severe depression/fatigue. The no fatigue/no depression group was chosen as the reference group. Finally, subgroup analyses by sex and age (cut-off median age = 70 years) were carried out.

All assumptions of multivariable linear regression models were ensured. The normal distribution of the residuals was visually assessed using Q-Q plots of standardized residuals. Cook’s D was calculated to investigate influential observations. We evaluated the homoscedasticity assumption by visually assessing the plots of predicted versus standardized residuals. The linearity assumptions between each continuous covariable in the model and the outcome were tested using the polynomial approach, where we included an additional squared term and considered the assumption violated if the corresponding *p*-value was below 0.05. There were no violations of the assumptions, except that the continuous fatigue and depression scores did not show linearity with the outcome. Therefore, the categorized variables were used in the regression analyses.

Furthermore, we checked for multicollinearity by calculating the variance inflation factor. No multicollinearity was present. The statistical analysis was performed based on a significance level of α = 0.05. All analyses were conducted using the Statistical Analysis System (SAS, version 9.4; SAS Institute Inc., Cary, NC, USA). 

## 3. Results

### 3.1. Descriptive Data

Of the 574 participants, 273 (47.6%) met the criteria for the presence of fatigue by the FAS score. About 16% of the included subjects showed moderate to severe depression due to the PHQ-9 questionnaire (cut-off ≥ 10 points). There were no sex differences regarding the prevalence of fatigue and depression. The means (±SD) of the total MacNew HRQoL score, the emotional domain, physical domain, and social domain of the study population were 5.44 (1.07), 5.37 (1.12), 5.47 (1.17), and 5.66 (1.14), respectively, for the total sample. Table 1 shows the sociodemographic and clinical characteristics of the study participants stratified by the presence of fatigue. AMI patients with fatigue were older, less often of German nationality, less educated, less often currently employed, had more often a history of diabetes or stroke, and had more frequently moderate to severe depression in comparison to patients without fatigue. 

The characteristics of patients according to the presence of depression measured by the PHQ-9 are presented in Table 2. AMI patients of German nationality suffered less often from depressive symptoms than non-German patients. Furthermore, patients with depression significantly more often reported fatigue than patients without depression. Differences between the groups were assessed by *t*-test in the case of continuous variables to compare the averages of two independent groups. The Chi-square test was conducted to examine whether or not there is a significant association between two categorical variables. 

### 3.2. Main Results

In Table 3, the results of the univariable associations between fatigue, depression, and the total score, as well as the three subscales of the MacNew HRQoL questionnaire, are presented. Both the presence of fatigue and depression were associated with a significantly reduced HRQoL (total score and all three subscales). Furthermore, we looked at a potential joint relationship between fatigue and depression and the total and all three subscales of the MacNew score. It was found that patients with fatigue without the presence of depression had a significantly reduced HRQoL in the total score (β-estimate −0.91; 95% CI −1.09–−0.77) and all three subscales of the MacNew questionnaire compared with patients without fatigue or depression. Depression without fatigue was stronger and significantly associated with the MacNew total score (β-estimate −2.15; 95% CI −2.77–−1.54) and all three subscales.

Table 4 presents the in multivariable regression analyses that found associations between fatigue, depression, the total score, and the subscales of the MacNew HRQoL questionnaire. Fatigue in AMI patients showed significant associations with the total MacNew score (β −1.15 (−1.30–−1.00), *p* < 0.0001), the physical (*p* < 0.0001), emotional (*p* < 0.0001), and social dimension (*p* < 0.0001) in linear regression models after adjustment for a variety of confounders. Also, depression was significantly associated with the total and all three subscales of the MacNew score (*p*-values < 0.0001 for each scale) in multivariable-adjusted linear regression models. 

The joint relationships between fatigue and depression and the total score and all three subscales of the MacNew score found in the univariable analyses could be confirmed even after adjustment for a number of confounding factors (Table 4).

### 3.3. Other Analyses

The results of the multivariable regression analyses found for the whole study population could be largely confirmed in subgroup analyses by sex and age (see Appendix A). Among men and women, both fatigue and depression, as well as joint relationships between fatigue and depression, were significantly associated with the total score and all three subscales of the MacNew HRQoL score (each *p*-value < 0.0001). In both age groups (<70/≥70 years), there was a significant association between fatigue and depression, respectively, and all scales of the McNew questionnaire after multivariable adjustment. The presence of depression in combination with fatigue was also significantly associated with all outcomes in both age groups. However, depression without concomitant fatigue showed a significant relationship with reduced quality of life in the total score and all three subscales only in the older age group.

## 4. Discussion

In the present study of AMI patients conducted during the COVID-19 pandemic, the frequencies of fatigue and moderate to severe depression assessed by patient-report (using validated questionnaires) were 47.6% and 16% two to 18 months after the acute event. Both fatigue and depression were associated with a decreased HRQoL, whereby a stronger impairment of HRQoL was shown in depressive patients. However, there was greater impairment when fatigue and depression were present together. This was true for the total score and the emotional, physical, and social dimensions of HRQoL. 

In previous studies, fatigue and depression have been associated with CHD and poor health status [38], whereby up to 76% of AMI patients experienced fatigue after the acute event [12,39]. Also, in a long-term study, 56% of the included AMI patients were found to suffer from fatigue for up to 6 years after an AMI [40]. Depression is also common in patients with heart failure (10 to 40%, depending on disease severity) and is associated with adverse clinical outcomes [41]. However, the symptom of fatigue is also common in the general population, with a prevalence ranging from 8% to 38% [42,43]. While some studies reported sex differences regarding fatigue scores, with women suffering from more severe fatigue [12], other investigations observed no sex differences [39,40]. In our study, the prevalence of fatigue was almost 48%, comparable to previous studies in myocardial infarction patients and higher than in the general population. 

Fatigue is a complex and multifaceted concept associated with, but distinct from, depression [44]. Symptoms of fatigue and depression overlap [45], but after an AMI, patients may also experience symptoms of fatigue without concurrent depression [16]. Prior studies have shown that depression is also common in CHD and MI patients [46]. Studies examining the prevalence of depression in CHD patients found that approximately 15% of patients were depressed [47], and a systematic review including 10,785 AMI survivors from 8 studies reported a weighted prevalence for all included studies of 20.5% [48]. A considerable proportion of AMI patients continued to be depressed in the year after discharge [48]. The prevalence of clinically relevant depressive symptoms in the general population ranges from 7% to 17% and is more common in certain populations, e.g., women, younger adults, and older adults [49,50]. With a frequency of approximately 10% of severe depression and 16% of patients with moderate to severe depression assessed by the PHQ-9, the results in our study are comparable to previous literature [47]. Furthermore, we could not observe significant differences in the prevalence of fatigue or depression between men and women after AMI. However, it is difficult to compare fatigue and depressive symptom scores between studies since different measures are used, or the symptoms are assessed at different time points after MI [47]. 

The present study showed that HRQoL (total, physical, emotional, and social subscale) was reduced but that the mean scores did not score lower compared to reference values after the AMI, as reported by Dixon et al. in 2002 [30]. Comparisons with previous literature and comparable studies on AMI patients before the pandemic are difficult, as similar studies are unavailable. A population-based German study covering roughly the same period showed that the actual HRQoL was slightly lower than German adult normative values from 2004 [51]. In line with other studies [52], one could conclude that the COVID-19 pandemic may not have manifested in substantial average HRQoL change or affected adult HRQoL levels equally. However, in this context, it could not be excluded that there may be subgroups vulnerable to HRQoL deterioration. 

In this investigation, fatigue and depression were associated with all three dimensions and the total score of HRQoL in multivariable regression analyses. Although patients with fatigue without concomitant depression also showed lower HRQoL in all subscales and total scores, HRQoL was comparatively reduced more in all dimensions in patients with depression. In another study, fatigue was also a predictor of decreased HRQoL one year after the acute heart attack [53]. 

Based on these results, in the treatment and care of post-AMI patients, it seems reasonable to identify patients with an increased risk of fatigue and depression to offer them appropriate support treatment and accompanying services. The present study was conducted during the COVID-19 pandemic, during which—after hospital discharge—cardiac rehabilitation services to improve the functional capacity, well-being and HRQoL of patients [54] could not be offered and provided, or only to a limited extent [55]. Furthermore, fatigue or even depressive mood often occurs in the context of post-COVID syndrome. In our study, very few patients (*n =* 38) had a previous infection with the SARS-CoV-2 virus, so it can be assumed that the symptoms and the reduced HRQoL reported in the study are not due to COVID-19. In addition, other pandemic-related factors that may have influenced the present results are conceivable. Due to the COVID-19 pandemic, local movement restrictions during lockdowns might have led to frustration, feelings of loneliness, and mental distress. Finally, fear of infection, financial losses, and interruptions in health care may have negatively impacted well-being and mental health. The extent to which the results found in our analyses apply to the non-pandemic period needs to be clarified in further studies on AMI patients.

### Strengths and Limitations

One of the strengths of the present study is the large number of well-characterized consecutive AMI patients from a population-based registry studied prospectively and the use of validated instruments. Limitations of the study include that no clinical diagnoses of “depressive disorder” and “fatigue syndrome” were available. In addition, no documentation on the functionality of the participants and the severity of the AMI was available in the study. Thus, stratified analyses due to AMI severity could not be conducted. Also, data on pandemic-related factors, e.g., the impact of lockdowns on patients with AMI, are missing. Furthermore, due to the cross-sectional association between fatigue, depression and HRQoL, a causal relationship cannot be deduced. The results may not be generalizable to other age groups or all ethnic groups since no information on ethnicity was available. 

## 5. Conclusions

In conclusion, we could show that fatigue and depression are negatively related to HRQoL in post-AMI patients. Both fatigue and depression, or a combination, were associated with a reduced overall HRQoL and the physical, emotional, and social dimensions of HRQoL. To protect mental health in future pandemics, healthcare providers need to pay more attention to mental health problems and underlying risk factors in the daily care of AMI patients. Therefore, it seems to make sense that physicians screen post-AMI patients regarding the presence of fatigue and depression to offer them appropriate support, therapy, and accompanying services to improve their HRQoL. 

## Figures and Tables

**Table 1 jcm-12-06349-t001:** Characteristics of the AMI patients (*n =* 574) included in the study by fatigue (yes/no).

	Fatigue No (*n =* 301)	Fatigue Yes (*n =* 273)	*p*-Value
Male sex	225 (74.6)	186 (68.1)	0.0790
Age (years)	67.9 (12.2)	70.0 (12.2)	0.0388
German nationality	286 (95.0)	245 (89.7)	0.0165
Secondary school	149 (49.5)	173 (63.4)	0.0008
Employment (yes)	103 (34.2)	68 (24.9)	0.0149
Living alone (*n =* 300/272)	65 (21.7)	66 (24.3)	0.4602
Hypertension	219 (72.8)	209 (76.6)	0.2966
Dyslipidemia	159 (52.8)	150 (55.0)	0.6107
Diabetes	71 (23.6)	90 (33.0)	0.0125
Prior stroke	16 (5.3)	30 (11.0)	0.0124
Prior infarction (*n =* 260/241 in no/yes fatigue groups)	35 (13.5)	44 (18.3)	0.1411
MI type:			0.8949
STEMI	127 (42.2)	112 (41.0)	
NSTEMI	133 (44.2)	118 (43.2)	
Bundle branch block (BBB)	27 (9.0)	27 (9.9)	
ECG not defined	14 (4.7)	16 (5.9)	
Prior SARS-CoV-2 infection (*n =* 295/264)	18 (6.1)	20 (7.6)	0.4894
BMI (kg/m^2^; *n =* 298/272)	27.8 (4.3)	27.4 (4.3)	0.3073
Smoking status (*n =* 299/273):			0.4512
Smoker	65 (21.6)	65 (23.8)	
Ex-Smoker	119 (39.5)	99 (36.3)	
Never smoker	115 (38.2)	109 (39.9)	
SARS-CoV-2 vaccination (*n =* 296/271)	277 (93.6)	253 (93.4)	0.9144
Any recanalisation therapy	282 (93.7)	249 (91.2)	0.2599
PHQ9 ≥ 10 points	6 (2.0)	86 (31.5)	<0.0001

**Table 2 jcm-12-06349-t002:** Characteristics of the AMI patients (*n =* 574) included in the study by depression (yes/no).

	Depression No (*n =* 482)	Depression Yes (*n =* 92)	*p*-Value
Male sex	351 (72.8)	60 (65.2)	0.1383
Age (years)	69.2 (12.2)	67.5 (12.2)	0.2161
German nationality	456 (94.6)	75 (81.5)	<0.0001
Secondary school	263 (54.6)	59 (64.1)	0.0902
Employment (yes)	144 (29.9)	27 (29.4)	0.9192
Living alone (*n =* 481/91 in no/yes depression groups)	108 (22.5)	23 (25.3)	0.5570
Hypertension	356 (73.9)	72 (78.3)	0.3743
Dyslipidemia	251 (52.1)	58 (63.0)	0.0531
Diabetes	128 (26.6)	33 (35.9)	0.0684
Prior stroke	37 (7.7)	9 (9.8)	0.4953
Prior infarction (*n =* 415/86)	61 (14.7)	18 (20.9)	0.1490
MI type:			0.7519
STEMI	202 (41.9)	37 (40.2)	
NSTEMI	209 (43.4)	42 (45.7)	
BBB	44 (9.1)	10 (10.9)	
ECG not defined	27 (5.6)	3 (3.3)	
Prior SARS-CoV-2 infection (*n =* 295/264)	31 (6.6)	7 (7.7)	0.7110
BMI (kg/m^2^; *n =* 478/92)	27.5 (4.3)	27.8 (4.5)	0.6348
Smoking status (*n =* 480/92):			0.8547
Smoker	107 (22.2)	23 (25.0)	
Ex-Smoker	185 (38.4)	33 (35.9)	
Never smoker	188 (39.0)	36 (39.1)	
SARS-CoV-2 vaccination (*n =* 475/92)	445 (93.7)	85 (92.4)	0.6458
Any recanalisation therapy	445 (92.3)	86 (93.5)	0.6999
Fatigue	187 (38.8)	86 (93.5)	<0.0001

**Table 3 jcm-12-06349-t003:** Results of the univariable linear regression models. Associations between fatigue and depression and HRQoL (MacNew total score and physical, emotional, and social dimensions).

	ß-Estimate (95% CI)	*p*-Value	R^2^
**Physical dimension**			
Fatigue	−1.31 (−1.47–−1.15)	<0.0001	0.3152
Depression	−1.67(−1.90–−1.45)	<0.0001	0.2804
No depression/no fatigue	Ref.		0.4366
No depression/fatigue	−1.03 (−1.20–−0.87)	<0.0001	
Depression/no fatigue	−2.31 (−3.03–−1.60)	<0.0001	
Depression and fatigue	−2.06 (−2.27–−1.85)	<0.0001	
**Emotional dimension**			
Fatigue	−1.28 (−1.43–−1.12)	<0.0001	0..3236
Depression	−1.92 (−2.12–−1.73)	<0.0001	0.4013
No depression/no fatigue	Ref.		0.5210
No depression/fatigue	−0.87 (−1.01–−0.72)	<0.0001	
Depression/no fatigue	−1.90 (−2.53–−1.27)	<0.0001	
Depression and fatigue	−2.29 (−2.48–−2.10)	<0.0001	
**Social dimension**			
Fatigue	−1.24 (−1.40–−1.08)	<0.0001	0.2942
Depression	−1.71 (−1.93–−1.50)	<0.0001	0.3077
No depression/no fatigue	Ref.		0.4413
No depression/fatigue	−0.93 (−1.09–−0.77)	<0.0001	
Depression/no fatigue	−2.41 (−3.10–−1.72)	<0.0001	
Depression and fatigue	−2.05 (−2.26–−1.85)	<0.0001	
**Total score**			
Fatigue	−1.24 (−1.39–−1.10)	<0.0001	0.3350
Depression	−1.74 (−1.93–−1.55)	<0.0001	0.3619
No depression/no fatigue	Ref.		0.5049
No depression/fatigue	−0.91 (−1.05–−0.77)	<0.0001	
Depression/no fatigue	−2.15 (−2.77–−1.54)	<0.0001	
Depression and fatigue	−2.09 (−2.27–−1.91)	<0.0001	

**Table 4 jcm-12-06349-t004:** Results of the multivariable linear regression models. Associations between fatigue and depression and HRQoL (MacNew total score and physical, emotional, and social dimensions).

	ß-Estimate (95% CI)	*p*-Value	R^2^
**Physical dimension**			
Fatigue	−1.19 (−1.35–−1.03)	<0.0001	0.3793
Depression	−1.61 (−1.83–−1.39)	<0.0001	0.3800
No depression/no fatigue	Ref.		0.4962
No depression/fatigue	−0.92 (−1.08–−0.75)	<0.0001	
Depression/no fatigue	−2.18 (−2.87–−1.48)	<0.0001	
Dpression and fatigue	−1.96 (−2.18–−1.75)	<0.0001	
**Emotional dimension**			
Fatigue	−1.21 (−1.37–−1.05)	<0.0001	0.3632
Depression	−1.87 (−2.07–−1.67)	<0.0001	0.4466
No depression/no fatigue	Ref.		0.5478
No depression/fatigue	−0.82 (−0.97–−0.68)	<0.0001	
Depression/no fatigue	−1.98 (−2.61–−1.34)	<0.0001	
Depression and fatigue	−2.22 (−2.42–−2.02)	<0.0001	
**Social dimension**			
Fatigue	−1.13 (−1.29–−0.96)	<0.0001	0.3462
Depression	−1.65 (−1.86–−1.43)	<0.0001	0.3861
No depression/no fatigue	Ref.		0.4859
No depression/fatigue	−0.83 (−0.99–−0.67)	<0.0001	
Depression/no fatigue	−2.30 (−2.98–−1.61)	<0.0001	
Depression and fatigue	−1.96 (−2.17–−1.75)	<0.0001	
**Total score**			
Fatigue	−1.15 (−1.29–−1.00)	<0.0001	0.3800
Depression	−1.68 (−1.87–−1.49)	<0.0001	0.4293
No depression/no fatigue	Ref.		0.5414
No depression/fatigue	−0.83 (−0.97–−0.68)	<0.0001	
Depression/no fatigue	−2.12 (−2.73–−1.51)	<0.0001	
Depression and fatigue	−2.01 (−2.20–−1.82)	<0.0001	

Adjusted for age, sex, diabetes, prior stroke, German nationality, highest school education, living alone, smoking status, BMI, any recanalisation therapy, hypertension, STEMI infarction, employment status, and prior SARS-CoV-2 infection.

## Data Availability

The data that support the findings of this study are available from the Chair of Epidemiology, Medical Faculty, University of Augsburg, but restrictions apply to the availability of these data, which are not publicly available. Data are, however, available from the authors upon reasonable request and with permission of the Chair of Epidemiology.

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
