# Peer review of "Fatigue, Depression and Health-Related Quality of Life in Patients with Post-Myocardial Infarction during the COVID-19 Pandemic: Results from the Augsburg Myocardial Infarction Registry"

_jcm, 2023, doi:10.3390/jcm12196349_

Round 1
Reviewer 1 Report
The abstract did not include statistical significance (for the most important results).
The introduction is very briefly written and should be expanded.
The exclusion criteria were not described.
For nonparametric equivalents of statistical tests, additional descriptive statistics (median, etc.) should be provided.
“All assumptions of multivariable linear regression models were ensured.” - it is worth describing what these assumptions are and pointing out their results.
Please provide detailed information on SAS software.
In the results, please indicate where and what statistical test was used.
The results for multivariate and univariate regression analysis should be presented and interpreted.
The limitations have been briefly described.
Minor editing of English language required.
Author Response
Thank you for evaluating our manuscript. All mentioned lines in our answers to your comments belong to the version with track changes.
The abstract did not include statistical significance (for the most important results).
Answer: Thank you for this hint. We revised the abstract accordingly.
The introduction is very briefly written and should be expanded.
Answer: As suggested by the reviewer, we expanded the introduction (see lines 59-66).
The exclusion criteria were not described.
Answer: Thank you for this hint. There were no exclusion criteria. All AMI cases admitted to the hospital between June 1st, 2020, and September 15th, 2021, and still alive were included in the study (see lines 79-84).
For nonparametric equivalents of statistical tests, additional descriptive statistics (median, etc.) should be provided.
Answer: Due to the suggestion of reviewer 2 the variable depression was now categorized into two categories instead of 4 categories. Thus, no non-parametric tests were carried out in the revision, but Chi-square tests for categorized variables and t-tests for continuous variables (see lines 145-148).
“All assumptions of multivariable linear regression models were ensured.” - it is worth describing what these assumptions are and pointing out their results.
Answer: As suggested, we described the assumptions and their results in the statistical analysis section (see lines 165-176). Normal distribution of the residuals was visually assessed using Q-Q plots of standardized residuals. Cook’s D was calculated to investigate influential observations. We evaluated the homoscedasticity assumption by visually assessing the plots of predicted versus standardized residuals. The linearity assumptions between each continuous covariate in the model and the outcome was tested using the polynomial approach, where we included an additional squared term and considered the assumption as violated if the corresponding p-value was below 0.05. there were no violations of the assumptions, except that the continuous fatigue and depression scores did not show linearity with the outcome. Therefore, the categorized variables were used in the regression analyses.
Please provide detailed information on SAS software.
Answer: Now we provide detailed information on SAS software (line 178).
In the results, please indicate where and what statistical test was used.
Answer: We added the statistical test used in the results section (see line 198-200).
The results for multivariate and univariate regression analysis should be presented and interpreted.
Answer: Thank you for this hint. We carried out new linear regression analyses using another cut-off of PHQ-9 as requested by reviewer 2 and present the results in a new Table 3 showing the results of the univariable analyses. Now in Table 4 the results of the new multivariable analyses requested by reviewer 2 are shown. We revised the methods and results sections accordingly (see line 148, lines 230-240, 244-254, and Tables 3 and 4).
The limitations have been briefly described.
Answer: Thank you for this comment. We have added some more limitations in the limitations section (lines 349-353).
Reviewer 2 Report
Thank you for inviting me to review this manuscript (“Fatigue, depression, and health-related quality of life in patients with post-myocardial infarction patients during a global epidemic”). The authors examined the relationship between fatigue, depression, and health-related quality of life (HRQoL) in patients who had an acute myocardial infarction (AMI) during COVID-19. The authors focused on AMI survivors derived from the Augsburg Myocardial Infarction Registry who had their event between June 1, 2020 and September 15, 2021. The outcome measures were the Fatigue Assessment Scale, Patient Health Questionnaire, and MacNew Heart Disease HRQoL questionnaire. The study suggested that the co-occurrence of both fatigue and depression was associated with a significantly reduced HRQoL. In summary, this study is worthwhile. Some comments for the authors to consider are detailed below.
TITLE
Consider including the 'COVID-19 pandemic' rather than the global epidemic.
The participants were derived from the Augsburg Myocardial Infarction Registry. It might be nice to acknowledge this in the title, as has been done elsewhere from this registry (e.g. PMCID: PMC5525217)
I think this title is better to narrative along these lines: 'Fatigue, depression, and health-related quality of life in patients with post-myocardial infarction during a global epidemic'.
INTRODUCTION
1. I suggest that after this sentence (“So far, the interplay between fatigue and depression and their association with HRQoL after AMI received little attention and, to our knowledge, has not been investigated, especially during the COVID-19 epidemic”.), the authors could give a rationale why to focus on the COVID-19 epidemic. Both psychosocial variables (fatigue and depression) would be affected by the curfew and physical distancing imposed during the pandemic. In general, some rationale is needed as to why this study was conducted during this particular period.
METHOD
1. Unlike FAS where the authors presented the cutoff point employed (“In the present analysis, individuals with scores of 22 and above were classified as fatigue”), however, for PHQ-9, the authors relied on an unnecessary and convoluted approach, reporting the mood status of the participants in “moderate (10-14), moderately severe (15-19), and severe (20 to 27)”. It would be better to use an established cut-off point for PHQ-9. Interestingly, the cut-off point for PHQ-9 has been widely established (Gen Hosp Psychiatry. 2013 Sep-Oct;35(5):551-5. doi: 10.1016/j.genhosppsych.2013.04.006.)
2. Study population, why inclusion criteria were stipulated to be at least 25 years old. Consider giving the rationale for such a cutoff point.
3. This sentence [“The study complies with the Declaration of Helsinki. All study participants gave their informed consent in writing and the study was approved by the Ethics Committee of the Bavarian Medical Association (Bayerische Landesärztekammer; No 12057).”] could have its own subheading (e.g., ethical consideration).
4. One of the missing links in the manuscript that could increase the scientific merit of the work is the lack of documentation on the functionality and severity of the participants. For example, the severity of myocardial infarction can also be classified into different types according to the consensus of the Third Universal Definition of Myocardial Infarction as expounded by the Task Force of the American College of Cardiology/American Heart Association on Performance Measures (J Am Coll Cardiol. 2017 Oct 17;70(16):2048-2090. doi: 10.1016/j.jacc.2017.06.032. Epub 2017 Sep 21). Consider this point if possible. Otherwise, it should be highlighted as one of the limitations of this study.
5. The method could be improved if the authors adhere to STROBE (Strengthening the reporting of observational studies in epidemiology). Please, get one that matches your study from here (https://www.strobe-statement.org/checklists/). Introduce all the missing subheadings.
RESULT
1. Consider conducting a subgroup analysis to investigate whether the impact of fatigue and depression on HRQoL varies based on factors such as age, sex, comorbidities, or severity of AMI.
2. In general, your approach to statistical analysis is well structured and follows standard procedures for analyzing data. But I have some points for the authors to consider. The authors used the Kruskal-Wallis rank sum test for continuous variables and the Chi-square test for categorical variables to test differences between depression categories. It would be helpful to briefly explain why you chose these specific tests. This helps the readers understand the constraints intimately linked to your data.
3. Second, the authors appear to have used multivariate linear regression. It might be beneficial to consider adding a statement on how model fit or goodness-of-fit was assessed (e.g., R-squared, residual analysis) to ensure that the models adequately capture the relationships being investigated.
4. Regarding the covariates in your regression models based on a literature review, provide a brief rationale for why each of these covariates was included, which can enhance the robustness of your analysis.
5. Consider checking for multicollinearity among these covariates.
DISCUSSION
1. You mention that the study was carried out during the COVID-19 pandemic, which is important. However, you could dig a bit deeper into how the pandemic might have influenced your results. Did the pandemic affect the prevalence of fatigue, depression, or HRQoL differently than what could be expected in a nonpandemic period? Discuss possible pandemic-related factors that could have influenced your findings. As mentioned earlier, although the study was designed to investigate the impact of COVID-19 on patients after MIA, it did not explore specific factors related to COVID-19, such as the effects of lockdowns, fear of infection, or disruptions in healthcare services. This should be acknowledged, or this topic brought up as a limitation.
2. Although the authors discuss the clinical implications of their findings, they could elaborate on the practical implications for healthcare providers. What can they learn from your study to better care for patients with AMI, especially those at risk of fatigue and depression?
LANGUAGE AND CLARITY
Although I am not a native speaker, to me, some sentences are quite long and complex, which can make them difficult to follow.
REFERENCES
The authors employed 53 references. Most of them are recent and relevant.
Author Response
Thank you for inviting me to review this manuscript (“Fatigue, depression, and health-related quality of life in patients with post-myocardial infarction patients during a global epidemic”). The authors examined the relationship between fatigue, depression, and health-related quality of life (HRQoL) in patients who had an acute myocardial infarction (AMI) during COVID-19. The authors focused on AMI survivors derived from the Augsburg Myocardial Infarction Registry who had their event between June 1, 2020 and September 15, 2021. The outcome measures were the Fatigue Assessment Scale, Patient Health Questionnaire, and MacNew Heart Disease HRQoL questionnaire. The study suggested that the co-occurrence of both fatigue and depression was associated with a significantly reduced HRQoL. In summary, this study is worthwhile. Some comments for the authors to consider are detailed below.
First of all, we want to thank the reviewer for his/her evaluation of our manuscript and the positive feedback! All mentioned lines in our answers to your comments belong to the version with track changes.
TITLE
Consider including the 'COVID-19 pandemic' rather than the global epidemic.
Answer: As suggested by the reviewer, we included COVID-19 pandemic in the title.
The participants were derived from the Augsburg Myocardial Infarction Registry. It might be nice to acknowledge this in the title, as has been done elsewhere from this registry (e.g. PMCID: PMC5525217)
Answer: As suggested by the reviewer, we included „results from the Augsburg myocardial infarction registry“ in the title.
I think this title is better to narrative along these lines: 'Fatigue, depression, and health-related quality of life in patients with post-myocardial infarction during a global epidemic'.
Answer: Thank you. We also considered this suggestion in the title, which reads now as „Fatigue, depression and health-related quality of life in patients with post-myocardial infarction during the COVID-19 pandemic: results from the Augsburg myocardial infarction registry“.
INTRODUCTION
I suggest that after this sentence (“So far, the interplay between fatigue and depression and their association with HRQoL after AMI received little attention and, to our knowledge, has not been investigated, especially during the COVID-19 epidemic”.), the authors could give a rationale why to focus on the COVID-19 epidemic. Both psychosocial variables (fatigue and depression) would be affected by the curfew and physical distancing imposed during the pandemic. In general, some rationale is needed as to why this study was conducted during this particular period.
Answer: Thank you. We have added a paragraph to the introduction to give a rationale why we focused on the COVID-19 pandemic (see lines 59-66).
METHOD
Unlike FAS where the authors presented the cutoff point employed (“In the present analysis, individuals with scores of 22 and above were classified as fatigue”), however, for PHQ-9, the authors relied on an unnecessary and convoluted approach, reporting the mood status of the participants in “moderate (10-14), moderately severe (15-19), and severe (20 to 27)”. It would be better to use an established cut-off point for PHQ-9. Interestingly, the cut-off point for PHQ-9 has been widely established (Gen Hosp Psychiatry. 2013 Sep-Oct;35(5):551-5.doi: 10.1016/j.genhosppsych.2013.04.006.)
Answer: Thank you for this comment. Due to the suggestion of the reviewer we now used the cut-off value PHQ-9 ≥ 10 to define patients with moderate to severe depression. We calculated the Table 2 of the manuscript accordingly and revised the Results section. We cited the paper mentioned by the reviewer and added it to the References (Ref. #33). Furthermore, we carried out new linear multivariable regression analyses and present the results in a new Table 4. Furthermore, the results of the univariable analyses as requested by reviewer 1 are shown in Table 3. We revised the methods und results section accordingly (see lines 117-121, lines 195-200, 230-240, 244-254).
Study population, why inclusion criteria were stipulated to be at least 25 years old. Consider giving the rationale for such a cutoff point.
Answer: Thank you for this hint. The inclusion criteria for myocardial infarction registry were established as part of the WHO MONICA project, the precursor project of the Augsburg myocardial infarction registry. These inclusion criteria are still used today (see lines 81-82).
This sentence [“The study complies with the Declaration of Helsinki. All study participants gave their informed consent in writing and the study was approved by the Ethics Committee of the Bavarian Medical Association (Bayerische Landesärztekammer; No 12057).”] could have its own subheading (e.g., ethical consideration).
Answer: As suggested by the reviewer, we included a subheading „Ethical consideration“ (see line 93).
One of the missing links in the manuscript that could increase the scientific merit of the work is the lack of documentation on the functionality and severity of the participants. For example, the severity of myocardial infarction can also be classified into different types according to the consensus of the Third Universal Definition of Myocardial Infarction as expounded by the Task Force of the American College of Cardiology/American Heart Association on Performance Measures (J Am Coll Cardiol. 2017 Oct 17;70(16):2048-2090. doi: 10.1016/j.jacc.2017.06.032. Epub 2017 Sep 21). Consider this point if possible. Otherwise, it should be highlighted as one of the limitations of this study.
Answer: We agree with the reviewer that lack of documentation on the functionality of the participants and the severity of the myocardial infarctions is a shortcoming. We highlight this limitation in the limitations section (see lines 349-351).
The method could be improved if the authors adhere to STROBE (Strengthening the reporting of observational studies in epidemiology). Please, get one that matches your study from here (https://www.strobe-statement.org/checklists/). Introduce all the missing subheadings.
Answer: As suggested by the reviewer, we adhere to STROBE and submitted the form as supplementary material. Furthermore, we added subheadings.
RESULT
Consider conducting a subgroup analysis to investigate whether the impact of fatigue and depression on HRQoL varies based on factors such as age, sex, comorbidities, or severity of AMI.
Answer: We conducted sex- and age-stratified subanalyses to investigate the association of fatigue and depression on HRQoL.The results of these analyses can be found in the supplementary materials. Furthermore, we briefly mentioned these additional analyses in the methods (see lines 163-164) and results section (see lines 261-272). In the limitations section, we mention that no information on severity of AMI was available in the data set and thus, stratified analyses due to severity of AMI could not be conducted (see lines 351-353).
In general, your approach to statistical analysis is well structured and follows standard procedures for analyzing data. But I have some points for the authors to consider. The authors used the Kruskal-Wallis rank sum test for continuous variables and the Chi-square test for categorical variables to test differences between depression categories. It would be helpful to briefly explain why you chose these specific tests. This helps the readers understand the constraints intimately linked to your data.
Answer: We thank the reviewer for this comment. Due to his/her suggestion, we now categorized the PHQ-score into two categories (cut-off ≥ 10 points). Thus, the Kruskal-Wallis rank sum test was not used anymore. Now we used the t-test for comparison of continuous variables and the Chi² test for comparison of categorical variables. We revised the methods and results section accordingly (see lines 145-148).
Second, the authors appear to have used multivariate linear regression. It might be beneficial to consider adding a statement on how model fit or goodness-of-fit was assessed (e.g., R-squared, residual analysis) to ensure that the models adequately capture the relationships being investigated.
Answer: Thank you. We now have added the R-squared values for each regression analysis in Table 3 and Table 4.
Regarding the covariates in your regression models based on a literature review, provide a brief rationale for why each of these covariates was included, which can enhance the robustness of your analysis.
Answer: As covariables we included in our regression models variables, which are associated with the exposure or the outcome or both and which are not mediators (see lines 156-157 and reference 41).
Consider checking for multicollinearity among these covariates.
Answer: Thank you. We checked for multicollinearity by calculating the variance inflation factor. No multicollinearity was present (see lines 174-176).
DISCUSSION
You mention that the study was carried out during the COVID-19 pandemic, which is important. However, you could dig a bit deeper into how the pandemic might have influenced your results. Did the pandemic affect the prevalence of fatigue, depression, or HRQoL differently than what could be expected in a nonpandemic period? Discuss possible pandemic-related factors that could have influenced your findings. As mentioned earlier, although the study was designed to investigate the impact of COVID-19 on patients after MIA, it did not explore specific factors related to COVID-19, such as the effects of lockdowns, fear of infection, or disruptions in healthcare services. This should be acknowledged, or this topic brought up as a limitation.
Answer: Thank you for this comment. We have added some points in the discussion and acknowledged this shortcoming in the limitations (see lines 338-343, and lines 352-353).
Although the authors discuss the clinical implications of their findings, they could elaborate on the practical implications for healthcare providers. What can they learn from your study to better care for patients with AMI, especially those at risk of fatigue and depression?
Answer: As suggested by the reviewer, we added some thoughts regarding the practical implications for healthcare providers (lines 359-364).
LANGUAGE AND CLARITY
Although I am not a native speaker, to me, some sentences are quite long and complex, which can make them difficult to follow.
Answer: The manuscript was revised accordingly.
REFERENCES
The authors employed 53 references. Most of them are recent and relevant.
Answer: During the revision of the manuscript we added or deleted some references due to the requirements of the reviewers. Now, 59 references are included.
Round 2
Reviewer 1 Report
The article has been corrected accordingly.